# Relationships between Military Spending and Green Capital Formation: Complementary or Substitutes?

**Ramesh Chandra Das** * and **Imran Hussain**

Department of Economics, Vidyasagar University, Midnapore 721102, India;
tserrs_hussaini@mail.vidyasagar.ac.in
* Correspondence: rameshdas22@mail.vidyasagar.ac.in

**Abstract:** The world's so-called rich countries have still been spending a huge sum of their budgets on military heads, in spite of there being no such fears of multilateral formal wars. Further, there is no such strong evidence that military expenditures are capable of raising the GDPs of the concerned countries. The countries, as a result, have been squeezing allotments in their budgets upon real asset building spending such as on the social sectors and natural resource development. There is thus a trade-off between military spending and real asset building. The present study examines the long-run relationships with causal interplays between military spending and green capital and also identifies the crowds-in or crowds-out effects of military spending on green capital in the top 20 military power-owning countries for the period 1991–2020. The results show the existence of long-run relations between the two in the majority of the countries and military spending makes a cause to green capital in the long run. But, for a few countries, the study observes causal interplay between military heads and green capital heads. Finally, the study finds that the militarization practices crowd out the green capital formation in eight countries and the opposite outcome, the crowding-in effects, works in twelve countries.

**Keywords:** military spending; green capital; cointegration; causality; crowding-out; crowding-in

## 1. Introduction

There is an old debate about the growth impact of military expenditure in the so-called developed countries of the world which have a large military base. The protectionists believed that countries needed to secure their territories from foreign aggressions and thus should invest in developing their military base which would ultimately lead to peaceful domestic environments in which to do more economic activities. On the other hand, the anti-lobby, the pro-Keynesians, did not believe in any such myth, rather they claimed that it would lead to the retardation of income growths since those were the wastage of resources and strongly recommended in favour of economic spending by the governments, particularly when the global economy was under depressionary situations (Antonakis 1997; Chen 2014; Das et al. 2015; Das and Mukherjee 2018). Military expenditure is a wastage of resources, and it is a lavish expense when the cross-border tensions can be minimized by means of bilateral and multilateral talks. The debate becomes widespread when the developmental implications of military expenditure within a government's budget are considered. There is often a popular tagline in this respect, the guns–butter trade off. There are two strands in the literature about the debate: one confirms that military expenditure trades off with developmental expenditures like education and health expenditures and the other confirms that military expenditure and developmental expenditures are complementary. In the first one, there is a situation of crowding-out effects but in the second one, there is a crowding-in effect. According to Wang (2022) the general public and academia often hold the view that the military sector has a dominating crowding-out power over developmental programmes like education and healthcare which is partly due

to the painful recall of war-time rationing. In the case of a powerful crowding-out effect, a dividend of disarmament can be expected by reallocating part of the military budget to the social sectors' developments.

The pioneering work in this regard was by Russett (1969), who revealed that military spending in the United States crowded out education and healthcare expenditures. The study also held that similar findings were valid for Britain and France, but not for Canada. The results were also valid for the disaggregated data for the healthcare sector in the United States (Peroff 1976). Contrary to these results, studies such as Caputo (1975) reported a complementary relationship in which defense and healthcare spending in the US were found to be positively correlated. In the recent past, the studies on G7 (Zhang et al. 2017) and OECD (Lin et al. 2015) countries have provided evidence for a complementary relationship between defence and welfare expenditures using panel data. In the recent past, it has been observed, in countries like Australia and Sweden, that defence expenditure always maintains positive correlations with education and healthcare expenditures (Wang 2022). Very recently, Ikegami and Wang (2023) have found a significant crowding-out effect of military expenditure on domestic government health spending in 116 countries. The study further observed that, by interacting the military expenditure variable with income per capita, an increase in income per capita has the capacity to neutralize any such crowding-out effect of military expenditure on domestic government health spending. The less well-off countries stand to suffer most, and wealthy ones stand to suffer least, which justifies the existence of complementarity relations and substitute relations of different standards in the countries.

Besides the debates on the trade-off between guns and butter, there is another form of the same debate between guns and environmental damage and conservations. Some researchers, like Renner (1991) and Jorgenson et al. (2012), show that military activities lead to the generation of more pollutants, leading to more environmental degradation. This is the complementary relation between the two. However, the governments of the polluting countries sometimes make budgetary allocations to the activities related to the conservation of the nature. Hence, the guns vis-a-vis butter debate in terms of spending on environmental conservations again may hold, both in the crowding-in or crowding-out type situations. When all heads of the governments' budgets remain unchanged, and only the rise in military expenditure happens, then the governments can cut down the budgetary allocations to the heads of conservation of nature. On the other hand, with similar condition, rises in budgetary allocations to defence heads may also influence the governments to generate new funds for the conservation of nature. In the literature, the conservation capital investment is achieved by means of activities which lead to the generation of green capital. One of such form of green capital formation is the amount of research and development (R&D) expenditure on green technology invention (Guo et al. 2018; Shi and Yang 2022). The present study intervenes in this area and aims to investigate empirically whether budgetary allocations for the military spending in the top twenty countries of the world (in terms of military power), lead to crowding-out or crowding-in situations of their budgetary allocations to green capital formations, the real guns and butter relationships for the future generations.

In the standard macroeconomic system, the equilibrium in the commodity producing sector, keeping all the other sectors of the economy intact, is given by the following expression:

$$Y = C + I + G + NX \tag{1}$$

where Y is GDP, C is private consumption expenditure, I is private investment expenditure, G is budgetary expenditure upon consumption, military spending, and conservation capital expenditure, and NX is net export. All of the right-hand side components are dependent upon income (Y). Given the budgetary allocation of public consumption, the total G is now divided between military spending (M) and environmental capital formation I, i.e., $G = M(Y) + E(Y)$, where $m = \frac{dM}{dY}$, $e = \frac{dE}{dY}$ and $0 < m, e < 1$. Further, there is an interaction term between M and E which represents the substitute (crowding-out) or complementary (crowding-in) relationships. Hence, the new factor of aggregate demand is $U = M*E$. The

relationship is substitute when $\frac{dU}{dY} < 0$ and complement when $\frac{dU}{dY} > 0$. The equilibrium relation has now become:

$$Y = C(Y) + I(Y) + G(Y) + NX(Y) \tag{2}$$

where export is exogenous.

$$Y = C(Y) + I(Y) + G(Y) + NX(Y)$$
$$\text{Or, } Y = C(Y) + I(Y) + M(Y) + E(Y) + M*E(Y) + NX(Y) \tag{3}$$

Let us consider linear forms of the above components of aggregate demand, the expression for Y is now:

$$Y = C_0 + C_y Y + I_0 + I_y Y + M_0 + M_y Y + E_0 + E_y Y + M_0 E_0 + M_y E_y \cdot Y \tag{4}$$

Here the marginal values of C, I, M, E are positive and less than unitary, i.e., $0 < C_y, I_y, M_y, E_y > 0$. For the sake of simplicity, we have omitted the international transaction part from the Y expression.

Now equilibrium income is

$$Y = \frac{C_0 + I_0 + M_0 + E_0 + M_0 E_0}{1 - C_y - I_y - M_y - E_y - M_y E_y} \tag{5}$$

For stability reasons, $(C_y + I_y + M_y + E_y + M_y E_y) < 1$. But the magnitude of $(C_y + I_y + M_y + E_y + M_y E_y)$ will be greater when military expenditure trades off with green capital expenditure (i.e., $M_y E_y < 0$) and will be lesser when military expenditure trades in with green capital expenditure (i.e., $M_y E_y > 0$). On the other hand, the denominator will be relatively larger, and Y will be relatively smaller when a crowding-out effect is present and the denominator will be relatively smaller and Y will be relatively larger when a crowding-in effect is present.

Let us derive the expression for the multiplier with respect to any of the parameters. It is obtained that:

$$\frac{dY}{d(C_0 \text{ or } I_0 \text{ or } M_0 \text{ or } E_0)} = \frac{1}{1 - C_y - I_y - M_y - E_y - M_y E_y} > 0 \tag{6}$$

The sign of the multiplier will be still positive for the substitute relations since the interaction term should not outweigh the $(C_y + I_y + M_y + E_y)$ term as the countries' share of budgetary allocations to military and conservation capital is less than that of the allocations of households and firms' allotments for consumption and investment, respectively. Thus, we can say that the impacts of any parametric effects upon Y will be larger when military expenditure is complementary to green capital expenditure and, reversely, the impacts of any parametric effects upon Y will be less when military expenditure is substitute to green capital expenditure.

### 1.1. Review of Related Literature

There are a number of studies regarding the long-term controversy about whether the government should spend its money on 'welfare' for its population or 'warfare', which is money spent by the government for militarization, the 'guns-and-butter' argument. However, studies related to the crowding-in and crowding-out relationship between military expenditure and green capital are few in the extant literature. This study thus reviews the research works on the former areas to relate them with the present theme on military expenditure vis-à-vis green capital formation.

Many empirical studies have obtained the negative trade-off between defence expenditure and other social welfare expenditure using different countries and different types of data. Russett (1969) discovered a very strong negative association between military spending and government spending on health and education in the United States, France,

and the United Kingdom. Peroff (1976) investigated the conflict between defence and three social welfare policies—health, public assistance, and housing. The conflict was that defence spending was funded in part by reducing civilian spending on consumer and capital goods through higher-than-normal tax rates, and in part by reducing government spending on non-defence programmes. Dabelko and McCormick (1977) evaluated the negative impact of military spending on spending levels for public education and health through an opportunity–cost approach in which they found the presence of opportunity–cost in a number of countries during 1950–1972. Peroff and Podolak-Warren (1979) conducted empirical research on the potential influence of defence on the private health sector; the findings lend support to the idea of a trade-off. Deger (1985) suggested defence has a high physical-resource cost as well as extraordinarily high human-resource costs. Thus, developing countries can and should divert a small portion of their significant armament spending to human capital development. Further, in a study of 19 Latin American countries, Apostolakis (1992) confirmed a trade-off between military spending and spending on health, education, social security, and welfare in the region.

By contrast, some studies argue that military spending may render a positive effect on investment, economic growth, and welfare. Benoit (1973, 1978) showed, with the rank order correlation and regression analysis, that the countries with a heavy defence burden generally had the most rapid rate of non-defence output. Lindgren (1984) discovered empirical evidence showing the favourable implications of military spending in industrialised countries. In view of the Marxist impact, Baran and Sweezy (1968) regarded militarization as fundamental for the survival of capitalism. There is positive relationship between military expenditure economic growth (Ram 1995). Defence spending is thought to be beneficial to human capital formation since defence personnel and conscripts are physically well-trained and receive a good skills education. Military investment may also lead to technological advancements and even spin-offs in the defence area. As a result, complementarity between them has been found in the literature such that by Verner (1983), Harris et al. (1988), and Kollias and Paleologou (2011).

However, heavy militarisation can produce ecological problems via institutions, weapons, and the behaviour of military (Jorgenson and Clark 2011). Militarization adds to a variety of environmental pollutants. Atomic and nuclear bomb testing generate radioactive fallout that spreads around the planet (Commoner 1967, 1971; Jorgenson et al. 2012). The environmental repercussions of militarization are not restricted to war; large-scale conflicts occur even during times of peace (Jorgenson 2005; York 2008). Militarization causes further environmental pollution via energy consumption (Renner 1991; Hooks and Smith 2004, 2005; Clark et al. 2010). According to Rice (2007), $CO_2$ emissions are relatively higher in developed countries shaped by higher military expenditure. The empirical research by Bildirici (2017) exposed the unidirectional causality both from militarisation and energy consumption to $CO_2$. The similar causal relationship from military expenditure to $CO_2$ was examined by Gokmenoglu et al. (2021).

Sustainability has been a key topic in environmental discussions, with environmentalists emphasising its far-reaching implications. However, as affluence grows, so does ecological consciousness, making it the primary rationale for reducing environmental damage in the later stages of economic growth (Chen et al. 2020). The green economy is an approach for attaining long-term development (World Bank 2007). The green economy makes regular utilisation of energy resources to increase environmental performance while reducing climate risk (ESCAP 2012). It is crucial for incorporating inclusive environmental sustainability and global climate adaption into our domestic and global economic systems, while also ensuring a positive future for people and the environment (Guo et al. 2021). Modern technology for green economic development has to be employed to meet environmental sustainability targets (Abbas et al. 2020).

*1.2. Research Gaps and Objectives*

As we discussed in the preceding literature review, it is worth noting that the impact of militarization on economic growth and the environment is multifaceted. For sustainable economic growth, green capital should be adopted. The existing literature shows mixed economic impacts of military expenditure. For example, positive income and investment effects are found in the works of Benoit (1973, 1978); (Ram 1995); Verner (1983); Harris et al. (1988), etc., and the negative effects on the said variables are found in the works of Dabelko and McCormick (1977); Peroff and Podolak-Warren (1979); Deger (1985); Apostolakis (1992), etc. In addition, the literature shows the environmental costs of military expenditure such as in Jorgenson and Clark (2011); Rice (2007), etc. As far as R&D spending on green technology development and green capital formation is concerned, we have some good works such as those by Guo et al. (2018); Shi and Yang (2022), etc. However, we did not find any such studies in the literature which focused on examining the relationship between military expenditure and green capital formation in countries, groups, or regions. Since every country spends a significant amount of its income on militarisation, other forms of welfare spending are hampered. In this context, the present study frames objectives in order to deal with three research questions:

(a)   Is there any long-run and short trade-off between military expenditure and green capital formation in the economy?
(b)   Does militarization cause green capital or the reverse one?
(c)   Is there any crowding-in or crowding-out effects of military spending on green capital in the economy?

## 2. Materials and Methods

*2.1. Variables and the Data Source*

Military Expenditure (ME): The annual data of military expenditures are taken from the Stockholm International Peace Research Institute (SIPRI), an independent international institute in Sweden that conducts research on peace and conflict issues and is regarded as a trustworthy source in the defence literature. It includes all current and capital expenditures on the armed forces, including peacekeeping forces, defence ministries, and other government agencies involved in defence projects such as paramilitary forces deemed capable of conducting military operations, and military space activities. Military and civil personnel expenses include retirement pensions for military personnel and social services for personnel; operation and maintenance; procurement; military research and development; and military aid (in the donor country's military expenditures).

Green Capital (GC): This simply implies the environmentally related government research and development budget. The annual data are taken from the Organisation for Economic Co-operation and Development (OECD).

The present study is based on time series annual data from 1993 to 2018 for the top twenty military spending countries—the United States, China, India, the United Kingdom, Russia, France, Germany, Japan, Korea, Italy, Australia, Canada, Iran, Israel, Spain, Brazil, Turkiye, the Netherlands, Poland and Singapore.

*2.2. Hypotheses of the Study*

The present study aims to testing the following four hypotheses:

(a)   There is no long-run relationship between military expenditure and green capital.
(b)   There are no short-run dynamics between military expenditure and green capital.
(c)   There is no causal interplay between military expenditure and green capital.
(d)   There is no crowding-out effect of military expenditure on green capital.

*2.3. Methodology*

Dealing with the *first* and *second* hypotheses, the present study used time series econometric tools to examine the long-run association, short-run dynamics, and causal

relationship between military expenditure and green capital. At first, a unit root test was performed to avoid spurious regressions; an Engle-Granger cointegration test was exercised to examine the existence of a long-term stable relationship; an Error Correction Model (ECM) was estimated to show the short-run dynamics between them around the long-run equilibrium relationship; and then the Granger causality test was performed to identify the directions of causality between the two variables. Finally, for the testing of the *third* hypothesis, the study employed a method to examine the crowding-in or crowding-out effects of military expenditure on green capital simply by checking the positive and negative signs, respectively, of the expression representing the ratio of two elasticities, elasticity of income with respect to military expenditure, and elasticity of income with respect to green capital spending. In this regard, the study follows the work of Das et al. (2018).

### 2.3.1. Examinations of First, Second and Third Hypotheses
Unit Root Test

To avoid non-spurious estimation, the present study conducted unit root testing by applying both Augmented Dickey–Fuller (ADF, Dickey and Fuller 1979) and Phillips and Perron (PP, Phillips and Perron 1988) techniques with the null hypothesis of 'non-stationarity' for the variables. Here ADF is a parametric and PP is a non-parametric test.

The study used ADF model as:

$$\Delta(Z)_t = \alpha + \delta(Z)_{t-1} + \sum_{i=1}^{n} \gamma_i \Delta(Z)_{t-i} + u_t \tag{7}$$

and the PP-test regression as:

$$\Delta(Z)_t = \alpha + \delta(Z)_{t-1} + u_t \tag{8}$$

where $Z$ is any time series variable which is to be tested for non-stationarity; '$\delta$' is equal to $\rho - 1$, '$n$' is the time lag of that variable and $u_t$ is the white noise disturbance term. Here the null hypothesis $H_0$: $\delta = 0$ (non-stationary or, $\rho = 1$) was tested by using a $\tau$-statistic. If the variable takes 'm' times differencing to be stationary, the series is then integrated in the order 'm', i.e., I(m).

Engle-Granger Cointegration Test

Now, to examine the existence of long-run relationship between ME and GC, Engle and Granger (1987) cointegration test technique was applied. This is a simple two-step unit root method which shows that if the two variables are I(1) and the corresponding estimated residual series is I(0), the lower order integration, then it can be said to be cointegrated. This implies that a long-run relationship exists between these two variables. The estimated residual series is derived as:

$$\hat{u} = GC_t - \hat{GC}_t \tag{9}$$

where,

$$GC_t = a + b^* ME_t \tag{10}$$

To conclude, the study used ADF-test for the residual series (Equation (9)) in line with (Equation (7)) from these two time series variables across the nations.

Error Correction Model

The Error Correction Model (ECM) is estimated to show the short-run dynamics between ME and GC around the long-run equilibrium relationship between them. The model is:

$$\Delta GC_t = \mu_0 + \mu_1 \Delta ME_t - \mu_2 \hat{u}_{t-1} \tag{11}$$

where, $\mu_0$ is constant term in the model, $\mu_1$ denotes the short-term coefficient of ME, and $\mu_2$ implies the coefficient of Error Correction Term (ECT) and $\hat{u}_{t-1}$ is the one-year lag

ECT. The non-zero value of $\mu_2$ implies the existence of disequilibrium between these two variables in the short run. The disequilibrium will be corrected over time and the long-run stable relationship will restore if, and only if, the value of $\mu_2$ is negative and statistically significant. This also implies the long-run causal relationship that runs from military heads to green capital.

Granger Causality Test

Besides the long-run causality, the study has examined the short-run causal interplay between these two variables by applying the Granger causality test (Granger 1969). The test procedure follows, estimating the following pairs of equations:

$$\Delta(GC)_t = C_1 + \sum_{i=1}^{m} \alpha_i \Delta(ME)_{t-i} + \sum_{j=1}^{n} \beta_j \Delta(GC)_{t-j} + \lambda_{1t} \tag{12}$$

$$\Delta(ME)_t = C_2 + \sum_{i=1}^{m} \gamma_i \Delta(ME)_{t-i} + \sum_{j=1}^{n} \delta_j \Delta(GC)_{t-j} + \lambda_{2t} \tag{13}$$

where $\lambda_{1t}$ and $\lambda_{2t}$ are uncorrelated white noise random error terms, '*m*' and '*n*' are the appropriate lags of $\Delta ME$ and $\Delta GC$, respectively, in the model. Here the null hypotheses are:

$$H_{01}: \; \alpha_i = 0 \; \& \; \delta_j = 0 \; (i = 1, \ldots, m) \; [\text{ME does not Granger cause GC}] \tag{14}$$

$$H_{02}: \; \delta_j = 0 \; \& \; \alpha_i = 0 \; (j = 1, \ldots, n) \; [\text{GC does not Granger cause ME}] \tag{15}$$

The hypotheses are tested by applying the following test statistic:

$$F = \frac{(RSS_R - RSS_{UR})/m}{RSS_{UR}/(n-k)} \tag{16}$$

where, respectively, $RSS_R$ and $RSS_{UR}$ stand for restricted and unrestricted residual sum square and '*k*' denotes the number of explanatory variables. The individual rejection of hypotheses $H_{01}$ and $H_{02}$ implies that ME Granger causes GC, and GC Granger causes ME, respectively. This is the notion of unilateral causality. The simultaneous rejection of both null hypotheses suggests that there is a two-way or bilateral causal relationship between them. Again, the individual acceptance of hypotheses $H_{01}$ and $H_{02}$ implies that ME does not Granger cause GC and GC does not Granger cause ME, respectively, and it is termed as a no-way causal relationship between them.

2.3.2. Testing for the Fourth Hypothesis (Crowding-In and Crowding-Out Effects)

The presence of causality results from ME to GC motivates us to examine whether there are crowding-in or crowding-out effects between the two. Recall that a crowding-out effect means substitution relation and a crowding-in effect means complementary relations between ME and GC. The way to examine the crowding-in or crowding-out effects of military expenditure on green capital is simply by taking into consideration the theoretical model (Equations (7)–(12)) and checking the signs of the following expression:

$$CE_{ME \; on \; GC} = \frac{\eta_{ME}}{\eta_{GC}} = \frac{dGC}{dME} * \frac{ME}{GC} \tag{17}$$

where,

$$\eta_{GC} = \frac{dGDP}{dGC} * \frac{GC}{GDP} \tag{18}$$

$$\eta_{ME} = \frac{dGDP}{dME} * \frac{ME}{GDP} \tag{19}$$

where $CE_{ME \; on \; GC}$ represents the crowding effect of military expenditure on green capital and this long-term impact is via output expansion and contraction by these two variables, and $\eta_{GC}$ and $\eta_{ME}$ implies the elasticity of output with respect to GC and ME, respectively.

If the sign of $CE_{ME\ on\ GC}$ is positive then expenditure on military crowds-in the green capital (also known as complementary effects) in the long run, and if it is negative then military expenditure crowds-out the green capital (also known as substitution effects) in the long run. The crowding-in or crowding-out effects based on changes in the average values of the military and green investments are long-run dynamics, which should not have any conflict. The causality results give us the short-run dynamics among the variables. In other words, even if we find, for instance, a strong direction of causality from military to green investments in a nation, the crowding-in or crowding-out effects are not always ruled out. The same applies to nations in which there are no causal links between these two investment channels.

## 3. Results and Discussion

This study first presents the data on the two variables across the countries for the entire period to have a graphical view on the trends of the two series. Figure 1 shows this for the military expenditure and Figure 2 shows that of the green capital investment.

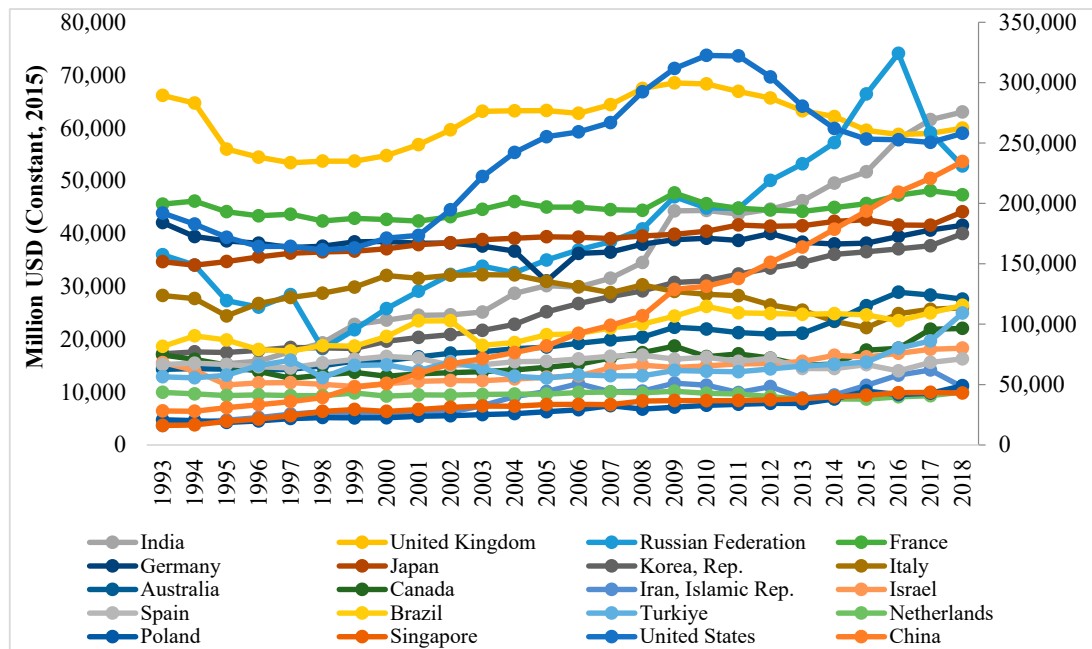

**Figure 1.** Military expenditure of the countries. Notes: In case of the United States, original data are divided by 2.5 for all the years to adjust the gap of military expenditure among the other nations. On the secondary axis, the values for US and China are measured to avoid clumsiness in the diagram. Source: Drawn by the authors.

It can be observed from Figure 1 that the amount of expenditure in the military head is rising for all the countries in the list even if there is no apprehension of formal wars. The USA tops the list followed by China and Singapore is at the 20th spot based on an average.

Figure 2 reveals that there are some countries which have steep rising trends in their green capital investments such as China and South Korea, Japan, etc. On the other hand, there are the countries like the USA, France, Spain, and the Netherlands where the trends are declining, particularly after the phase of global financial crisis. It may be that the rising atmospheric pollution in countries like China and South Korea has compelled these countries to invest more into green capital, and, on the other hand, the falling trends in the atmospheric pollution in the countries like the USA, France, Spain, and the Netherlands have incentivised them to cut down the amount of green investment.

Table 1 shows the average values of the two heads of budgetary expenses for the entire period across the countries. The USA leads the group in ME but China in GC. In the last two columns of the table the Pearson correlation coefficients between ME and GC are

given with their respective probabilities of the t statistics. In most of the countries the said correlation coefficients are positive and statistically significant. However, the way of the causations are not clear from the correlation analysis, and we need the time series analysis for the purpose which are attempted below.

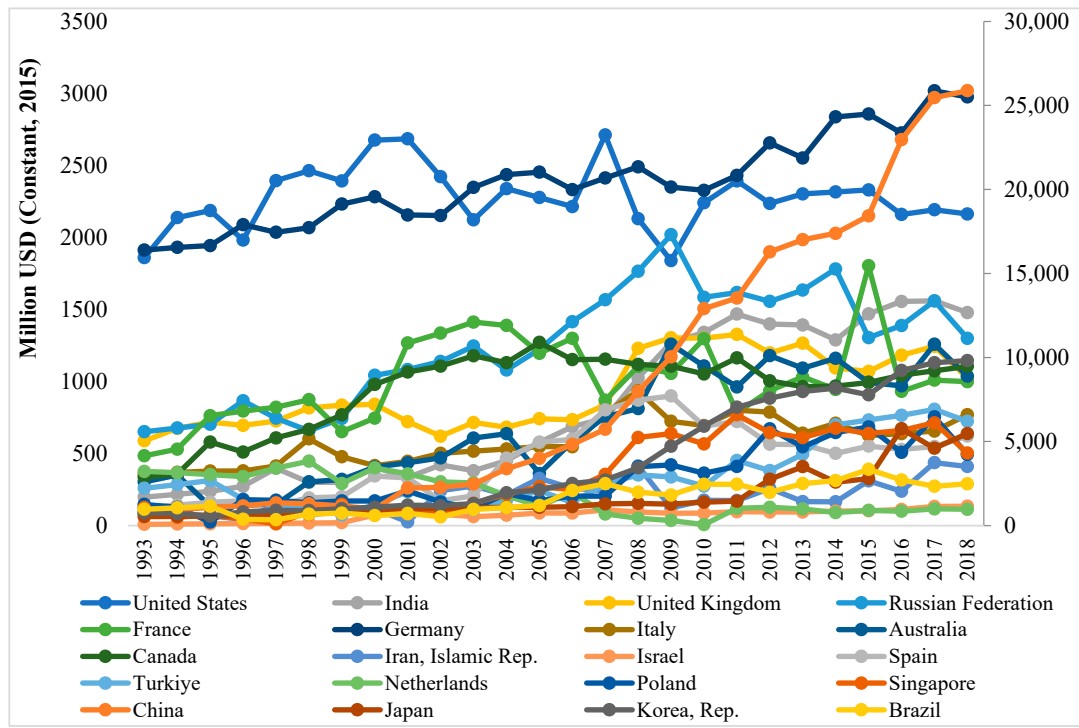

**Figure 2.** Trends of green capital investment of the countries. Notes: On the secondary axis the values for China, Japan, Brazil, and Korea are measured to avoid clumsiness in the diagram. Source: Drawn by the authors.

**Table 1.** Mean Value and the Correlation coefficient.

| Country | Mean Value (In Million USD, Constant 2015) | | Correlation Coefficient | |
| --- | --- | --- | --- | --- |
| | $\overline{ME}$ | $\overline{GC}$ | r | *p*-Value |
| United States | 586,162.53 | 2275.52 | −0.21 | 0.3151 |
| China | 103,792.45 | 8521.66 | 0.98 | 0.0000 |
| India | 33,851.69 | 814.04 | 0.96 | 0.0000 |
| United Kingdom | 61,148.89 | 930.18 | 0.43 | 0.0285 |
| Russian Federation | 40,241.37 | 1244.23 | 0.65 | 0.0003 |
| France | 44,854.74 | 1012.69 | 0.12 | 0.5739 |
| Germany | 38,296.23 | 2384.60 | 0.11 | 0.6047 |
| Japan | 39,008.86 | 1760.17 | 0.77 | 0.0000 |
| Korea, Rep. | 26,518.96 | 3888.15 | 0.97 | 0.0000 |
| Italy | 28,216.23 | 586.11 | −0.19 | 0.3509 |
| Australia | 19,561.80 | 684.13 | 0.86 | 0.0000 |
| Canada | 15,907.51 | 940.70 | 0.23 | 0.2651 |
| Iran, Islamic Rep. | 8642.18 | 199.21 | 0.71 | 0.0000 |
| Israel | 14,104.62 | 72.12 | 0.69 | 0.0001 |
| Spain | 15,661.90 | 449.79 | 0.46 | 0.0184 |
| Brazil | 22,103.93 | 1602.15 | 0.78 | 0.0000 |
| Turkiye | 14,895.13 | 343.56 | 0.62 | 0.0007 |
| Netherlands | 9494.92 | 213.82 | −0.03 | 0.8919 |
| Poland | 6760.38 | 322.31 | 0.86 | 0.0000 |
| Singapore | 7391.10 | 349.21 | 0.84 | 0.0000 |

Note: $\overline{ME} = \frac{1}{T}\sum_{t=1}^{T} ME_t$, $\overline{GC} = \frac{1}{T}\sum_{t=1}^{T} GC_t$ and 'r' is the Karl Pearson correlation coefficient between ME and GC. Source: Authors' own computations.



### 3.1. Unit Root Test Results

Table 2 shows the results of unit root for the ADF and PP tests following Equations (7) and (8). The results of this study demonstrate that neither of the variables is stationary in their levels.

**Table 2.** Unit Root Test Results.

| Countries | | ΔME | | ΔGC | | Remarks |
|---|---|---|---|---|---|---|
| | | t-Statistic | Prob. | t-Statistic | Prob. | |
| United States | ADF | −2.3304 | 0.0221 | −6.5280 | 0.0000 | Stationary |
| | PP | −1.9630 | 0.0506 | −10.5837 | 0.0000 | Stationary |
| China | ADF | −3.2818 | 0.0273 | −3.2166 | 0.0314 | Stationary |
| | PP | −3.2041 | 0.0322 | −3.2166 | 0.0314 | Stationary |
| India | ADF | −4.2010 | 0.0035 | −3.9350 | 0.0064 | Stationary |
| | PP | −4.1619 | 0.0038 | −3.9261 | 0.0065 | Stationary |
| United Kingdom | ADF | −2.8249 | 0.0697 | −3.8068 | 0.0086 | Stationary |
| | PP | −2.7465 | 0.0811 | −3.8114 | 0.0085 | Stationary |
| Russian Federation | ADF | −3.6263 | 0.0132 | −5.3863 | 0.0002 | Stationary |
| | PP | −3.7599 | 0.0095 | −5.4978 | 0.0002 | Stationary |
| France | ADF | −4.9917 | 0.0005 | −7.8520 | 0.0000 | Stationary |
| | PP | −4.9912 | 0.0005 | −8.6017 | 0.0000 | Stationary |
| Germany | ADF | −6.0615 | 0.0000 | −5.8719 | 0.0001 | Stationary |
| | PP | −6.0615 | 0.0000 | −11.8675 | 0.0000 | Stationary |
| Japan | ADF | −5.2893 | 0.0003 | −5.6605 | 0.0001 | Stationary * |
| | PP | −3.4864 | 0.0175 | −7.5915 | 0.0000 | Stationary * |
| Korea, Rep. | ADF | −2.9005 | 0.0601 | −3.2717 | 0.0279 | Stationary |
| | PP | −2.8249 | 0.0697 | −3.2717 | 0.0279 | Stationary |
| Italy | ADF | −4.2069 | 0.0034 | −5.2785 | 0.0003 | Stationary |
| | PP | −4.2069 | 0.0034 | −8.6753 | 0.0000 | Stationary |
| Australia | ADF | −4.1631 | 0.0044 | −6.1361 | 0.0000 | Stationary |
| | PP | −2.9916 | 0.0599 | −6.5173 | 0.0000 | Stationary |
| Canada | ADF | −4.1057 | 0.0043 | −5.2541 | 0.0003 | Stationary |
| | PP | −4.1157 | 0.0042 | −5.2487 | 0.0003 | Stationary |
| Iran, Islamic Rep. | ADF | −4.0792 | 0.0046 | −8.3603 | 0.0000 | Stationary |
| | PP | −3.6826 | 0.0113 | −9.2776 | 0.0000 | Stationary |
| Israel | ADF | −6.7006 | 0.0000 | −4.6816 | 0.0011 | Stationary |
| | PP | −3.9041 | 0.0068 | −4.7317 | 0.0010 | Stationary |
| Spain | ADF | −5.5074 | 0.0002 | −4.1983 | 0.0035 | Stationary |
| | PP | −5.5099 | 0.0002 | −4.1983 | 0.0035 | Stationary |
| Brazil | ADF | −4.8854 | 0.0008 | −4.8792 | 0.0008 | Stationary |
| | PP | −5.0056 | 0.0005 | −5.4758 | 0.0002 | Stationary |
| Turkiye | ADF | −4.3234 | 0.0116 | −4.5342 | 0.0093 | Stationary |
| | PP | −4.3234 | 0.0116 | −6.6847 | 0.0001 | Stationary |
| Netherlands | ADF | −2.0233 | 0.0434 | −6.7537 | 0.0000 | Stationary * |
| | PP | −4.5519 | 0.0001 | −6.7537 | 0.0000 | Stationary * |

**Table 2.** *Cont.*

| Countries | | ΔME | | ΔGC | | Remarks |
|---|---|---|---|---|---|---|
| | | **t-Statistic** | **Prob.** | **t-Statistic** | **Prob.** | |
| Poland | ADF | −5.5588 | 0.0002 | −8.3070 | 0.0000 | Stationary |
| | PP | −4.8219 | 0.0008 | −9.1262 | 0.0000 | Stationary |
| Singapore | ADF | −3.4258 | 0.0205 | −4.9408 | 0.0006 | Stationary |
| | PP | −3.8885 | 0.0071 | −4.9307 | 0.0006 | Stationary |

Note: 'Δ' implies the variables are in first order difference. '*'—Stationary in logarithm value. The results for USA, UK, Korea, the Netherlands are taken with the test condition of 'none'. Source: Authors' own estimations.

By using the first difference of both the variables, the findings demonstrate stationarity of these two variables at a significant level of one and five percent. For Japan and the Netherlands, the variables are in logarithm values. The study concludes that variables are I (1), i.e., both Military Expenditure (ME) and Green Capital (GC) are integrated of order one. These findings enable the application of cointegration and the Error-Correction Model (ECM), with the required lag structure.

### 3.2. Engle–Granger Cointegration Test Results

Table 3 shows the results of unit root of the derived residuals through the ADF test procedure following Equation (9).

**Table 3.** Unit Root Test results for the Residuals, $\hat{u}_t$ (in level value).

| Countries | ADF | Prob. | Integrating Order | Remarks |
|---|---|---|---|---|
| United States | −4.1023 | 0.0041 | I = (0) | Cointegrated |
| China | −2.0249 | 0.0431 | I = (0) | Cointegrated |
| India | −1.3009 | 0.1733 | I ≠ (0) | Not cointegrated |
| United Kingdom | −2.1907 | 0.0300 | I = (0) | Cointegrated |
| Russian Federation | −2.2898 | 0.0240 | I = (0) | Cointegrated |
| France | −3.6350 | 0.0123 | I = (0) | Cointegrated |
| Germany | −1.3320 | 0.0841 | I ≠ (0) | Not cointegrated |
| Japan | −0.4857 | 0.8772 | I ≠ (0) | Not cointegrated |
| Korea, Rep. | −2.1940 | 0.0299 | I = (0) | Cointegrated |
| Italy | −1.9302 | 0.0527 | I = (0) | Cointegrated |
| Australia | −2.2999 | 0.0234 | I = (0) | Cointegrated |
| Canada | −2.9581 | 0.0529 | I = (0) | Cointegrated |
| Iran, Islamic Rep. | −4.6717 | 0.0011 | I = (0) | Cointegrated |
| Israel | −2.1869 | 0.0309 | I = (0) | Cointegrated |
| Spain | −1.3286 | 0.1652 | I ≠ (0) | Not cointegrated |
| Brazil | −2.6712 | 0.0097 | I = (0) | Cointegrated |
| Turkiye | −1.9954 | 0.0459 | I = (0) | Cointegrated |
| Netherlands | −2.2899 | 0.0240 | I = (0) | Cointegrated |
| Poland | −3.7237 | 0.0100 | I = (0) | Cointegrated |
| Singapore | −1.9278 | 0.0529 | I = (0) | Cointegrated |

Notes: 'I = (0)' implies that $\hat{u}_t$ is stationary in level, whereas, 'I ≠ (0)' implies that $\hat{u}_t$ is not stationary in level. The results for China, UK, Russia, Korea, Italy, Australia, Canada, Israel, Brazil, Turkiye, Netherlands and Singapore are taken with the test configuration of 'none'. Source: Authors' estimations.

The ADF test statistic on $\hat{u}_t$ is found to be significant under the five per cent level in sixteen countries, namely, the United States, China, the United Kingdom, Russia, France, Korea, Australia, Canada, Iran, Israel, Spain, Brazil, Turkiye, the Netherlands, Poland, and Singapore. This implies $\hat{u}_t \sim I(0)$ in these nations. The results of this study demonstrate that both ME and GC have a long-run relationship between them in these sixteen nations. In contrast to this, we have not found any long-run relationship between these two variables in four of the countries, India, Germany, Japan, and Spain.

### 3.3. Error Correction Results

The estimated ECM results of the countries that have cointegration results are shown in Table 4 following Equation (11).

**Table 4.** Estimation of Error Correction Model (ECM).

| Countries | Estimated ECM | Remarks |
|---|---|---|
| United States | $\Delta\widehat{GC} = 21123924 + 0.0019\,\Delta\text{ME} - 0.7525\,\hat{u}_{t-1}$ <br> (0.5965) (0.5024) (0.0066) | Errors corrected with LR causal influence |
| China | $\Delta\widehat{GC} = 224000000 + 0.0934\,\Delta\text{ME} - 0.2760\,\hat{u}_{t-1}$ <br> (0.5479) (0.0206) (0.0498) | Errors corrected with LR causal influence |
| United Kingdom | $\Delta\widehat{GC} = 18019378 + 0.0067\,\Delta\text{ME} - 0.2131\,\hat{u}_{t-1}$ <br> (0.4308) (0.4863) (0.0408) | Errors corrected with LR causal influence |
| Russian Federation | $\Delta\widehat{GC} = 22407286 + 0.0074\,\Delta\text{ME} - 0.2099\,\hat{u}_{t-1}$ <br> (0.5724) (0.3633) (0.1583) | Errors corrected with LR causal influence |
| France | $\Delta\widehat{GC} = 20636005 + 0.0247\,\Delta\text{ME} - 0.6587\,\hat{u}_{t-1}$ <br> (0.7119) (0.6348) (0.0033) | Errors corrected with LR causal influence |
| Korea, Rep. | $\Delta\widehat{GC} = 313000000 + 0.0515\,\Delta\text{ME} - 0.2636\,\hat{u}_{t-1}$ <br> (0.0482) (0.7060) (0.0221) | Errors corrected with LR causal influence |
| Italy | $\Delta\widehat{GC} = 15326298 + 0.0014\,\Delta\text{ME} - 0.2126\,\hat{u}_{t-1}$ <br> (0.4430) (0.9197) (0.1205) | Errors not corrected with no LR causal influence |
| Australia | $\Delta\widehat{GC} = 14379333 + 0.0344\,\Delta\text{ME} - 0.3694\,\hat{u}_{t-1}$ <br> (0.7003) (0.3240) (0.0435) | Errors corrected with LR causal influence |
| Canada | $\Delta\widehat{GC} = 27691775 + 0.0148\,\Delta\text{ME} - 0.1730\,\hat{u}_{t-1}$ <br> (0.1192) (0.3358) (0.0225) | Errors corrected with LR causal influence |
| Iran, Islamic Rep. | $\Delta\widehat{GC} = 2793689 + 0.0157\,\Delta\text{ME} - 0.9975\,\hat{u}_{t-1}$ <br> (0.8746) (0.2722) (0.0003) | Errors corrected with LR causal influence |
| Israel | $\Delta\widehat{GC} = 3618961 + 0.0115\,\Delta\text{ME} - 0.2725\,\hat{u}_{t-1}$ <br> (0.1511) (0.0092) (0.0259) | Errors corrected with LR causal influence |
| Brazil | $\Delta\widehat{GC} = 46618704 + 0.0523\,\Delta\text{ME} - 0.2955\,\hat{u}_{t-1}$ <br> (0.5736) (0.3551) (0.0610) | Errors corrected with LR causal influence |
| Turkiye | $\Delta\widehat{GC} = 30704299 + 0.0267\,\Delta\text{ME} + 0.1303\,\hat{u}_{t-1}$ <br> (0.0926) (0.0516) (0.2815) | Errors not corrected with no LR causal influence |
| Netherlands | $\Delta\widehat{GC} = -0.045519 - 0.4688\,\Delta\text{ME} - 0.3362\,\hat{u}_{t-1}$ <br> (0.7566) (0.9136) (0.0376) | Errors corrected with LR causal influence |
| Poland | $\Delta\widehat{GC} = 13044321 + 0.0319\,\Delta\text{ME} - 0.7525\,\hat{u}_{t-1}$ <br> (0.5965) (0.5024) (0.0066) | Errors corrected with LR causal influence |
| Singapore | $\Delta\widehat{GC} = -12519605 + 0.1180\,\Delta\text{ME} - 0.2248\,\hat{u}_{t-1}$ <br> (0.5955) (0.0719) (0.0793) | Errors corrected with LR causal influence |

Note: '$\Delta$' denotes the first difference operator and $\widehat{GC}$ implies the estimated value of GC. The corresponding probability values are in the first bracket; LR causality: Long-run causality runs from ME to GC. Source: Authors' estimations.

In Table 4, the coefficient of $\hat{u}_{t-1}$ is found to be negative and significant in fourteen countries—the United States, China, the United Kingdom, Russia, France, Korea, Australia, Canada, Iran, Israel, Brazil, the Netherlands, Poland and Singapore. This means that if there is a deviation from the long-term stable position, the errors are corrected instantaneously and the long-term equilibrium relation is re-established. The long-run causal relationship is thus running from ME to GC in these fourteen countries. It implies that, in the long-run, green capital is significantly affected by militarization in these nations. On the other hand,

the study finds no long-run causal relationship from ME to GC in the cases of Italy and Turkiye only.

### 3.4. Granger Causality Test Results

The Granger causality test results are shown in Table 5 following the set of Equations (12) and (13).

**Table 5.** Granger causality test results.

| Countries | Null EE↛ Hypothesis | Lag | F-Stat | Prob. | Remarks |
|---|---|---|---|---|---|
| United States | D(ME) ↛ D(GC)<br>D(GC) ↛ D(ME) | 2 | 2.6325<br>0.1296 | 0.0993<br>0.8792 | No SR causality |
| China | D(GC) ↛ D(ME)<br>D(ME) ↛ D(GC) | 1 | 4.9010<br>1.4012 | 0.0380<br>0.2498 | GC → ME |
| India | D(GC) ↛ D(ME)<br>D(ME) ↛ D(GC) | 2 | 1.3699<br>0.0806 | 0.2794<br>0.9229 | No SR causality |
| United Kingdom | D(GC) ↛ D(ME)<br>D(ME) ↛ D(GC) | 2 | 0.1758<br>0.2752 | 0.8402<br>0.7626 | No SR causality |
| Russian Federation | D(GC) ↛ D(ME)<br>D(ME) ↛ D(GC) | 2 | 0.4358<br>0.7491 | 0.6334<br>0.4870 | No SR causality |
| France | D(GC) ↛ D(ME)<br>D(ME) ↛ D(GC) | 2 | 0.7691<br>0.8419 | 0.4781<br>0.4472 | No SR causality |
| Germany | D(GC) ↛ D(ME)<br>D(ME) ↛ D(GC) | 2 | 1.1909<br>0.0312 | 0.3268<br>0.9494 | No SR causality |
| Japan | D(lnGC) ↛ D(lnME)<br>D(lnME) ↛ D(lnGC) | 2 | 3.4387<br>0.3120 | 0.0544<br>0.7359 | GC → ME |
| Korea, Rep. | D(ME) ↛ D(GC)<br>D(GC) ↛ D(ME) | 4 | 3.6662<br>0.3198 | 0.0357<br>0.8593 | ME → GC |
| Italy | D(GC) ↛ D(ME)<br>D(ME) ↛ D(GC) | 2 | 0.1702<br>0.1207 | 0.8448<br>0.8870 | No SR causality |
| Australia | D(ME) ↛ D(GC)<br>D(GC) ↛ D(ME) | 2 | 0.0088<br>1.4155 | 0.9912<br>0.2686 | No SR causality |
| Canada | D(ME) ↛ D(GC)<br>D(GC) ↛ D(ME) | 2 | 0.3748<br>0.3655 | 0.6927<br>0.6989 | No SR causality |
| Iran, Islamic Rep. | D(ME) ↛ D(GC)<br>D(GC) ↛ D(ME) | 2 | 0.6242<br>0.5720 | 0.5469<br>0.5743 | No SR causality |
| Israel | D(ME) ↛ D(GC)<br>D(GC) ↛ D(ME) | 2 | 0.3687<br>0.1647 | 0.6968<br>0.8494 | No SR causality |
| Spain | D(GC) ↛ D(ME)<br>D(ME) ↛ D(GC) | 1 | 1.6238<br>4.1303 | 0.2165<br>0.0550 | ME → GC |
| Brazil | D(ME) ↛ D(GC)<br>D(GC) ↛ D(ME) | 2 | 0.4873<br>0.0753 | 0.6221<br>0.9277 | No SR causality |
| Turkiye | D(GC) ↛ D(ME)<br>D(ME) ↛ D(GC) | 2 | 3.2562<br>0.0216 | 0.0611<br>0.9789 | GC → ME |
| Netherlands | D(lnME) ↛ D(lnGC)<br>D(lnGC) ↛ D(lnME) | 2 | 1.2395<br>0.6073 | 0.3131<br>0.5556 | No SR causality |
| Poland | D(ME) ↛ D(GC)<br>D(GC) ↛ D(ME) | 2 | 1.4841<br>0.6885 | 0.2532<br>0.5151 | No SR causality |
| Singapore | D(ME) ↛ D(GC)<br>D(GC) ↛ D(ME) | 2 | 0.3920<br>2.0885 | 0.6814<br>0.1529 | No SR causality |

Note: '↛' implies does not Granger cause; '→' implies Granger cause; Lag values are justified by Akaike Information Criterion (AIC). Source: Authors' estimation.

As shown in Table 5, there is the presence of short-run causal interplays between ME and GC in five countries—China, Japan, Korea, Spain and Turkiye. In the case of Korea and Spain, the causal relation is running from ME to GC both in the short run as well as long-run. Whereas, in the short run, the causal relations from GC to ME are found in China, Japan, and Turkiye. No short-run causal relationship was found in the fifteen remaining countries.

Thus, it is evident from Tables 4 and 5 that few countries have a causal interplay from ME to GC in the short term, but the study has observed that this causal relationship exists in a considerable number of nations over the long term.

### 3.5. Results of Crowding-In and Crowding-Out Effects

The results of crowding-in and crowding-out effects are shown in Table 6 following Equation (17).

**Table 6.** Crowding-in and Crowding-out Effect of ME on GC.

| Countries | $CE_{MEonGC} = \frac{dGC}{dME} * \frac{ME}{GC}$ | Sign | Remarks |
|---|---|---|---|
| United States | 14.042 | + | Crowding-in |
| China | 1.1369 | + | Crowding-in |
| India | 4.6227 | + | Crowding-in |
| United Kingdom | −88.591 | − | Crowding-out |
| Russian Federation | 0.0646 | + | Crowding-in |
| France | −37.696 | − | Crowding-out |
| Germany | −0.0986 | − | Crowding-out |
| Japan | 6.4122 | + | Crowding-in |
| Korea, Rep. | 4.4175 | + | Crowding-in |
| Italy | 0.2448 | + | Crowding-in |
| Australia | 6.4760 | + | Crowding-in |
| Canada | 0.5034 | + | Crowding-in |
| Iran, Islamic Rep. | −10.215 | − | Crowding-out |
| Israel | 6.7788 | + | Crowding-in |
| Spain | 13.253 | + | Crowding-in |
| Brazil | −9.2422 | − | Crowding-out |
| Turkiye | −26.945 | − | Crowding-out |
| Netherlands | −50.748 | − | Crowding-out |
| Poland | 7.9090 | + | Crowding-in |
| Singapore | −0.2469 | − | Crowding-out |

Source: Authors' estimation.

There are mixed cases of results in favour of both crowding-in as well as crowding-out effects from military expenditure to green capital over time taken for the study. There are eight countries in which we observe a crowding-out effect of ME on GC (five countries belong to high-income group and the remaining three countries are under the middle-income group of the world). These countries are the United Kingdom, France, Germany, Iran, Brazil, Turkiye, the Netherlands and Singapore. The rise in military spending has led to reduced expenditures on green capital. The results support the 'Gun–Butter' argument and the clarification is intuitive in the sense that, in the presence of dynamic spillovers from a particular government budget, an increase in military expenditures will drive out an equivalent amount of all other social spending (Domke et al. 1983; Scheetz 1992; Yildirim and Sezgin 2002).

By contrast, the study has found crowding-in effects in the remaining twelve countries (where eight countries belong to high-income group and four countries belong to middle-income group of the world)—the United States, China, India, Russia, Japan, Korea, Italy, Australia, Canada, Israel, Spain and Poland. This result implies a trade-in relationship between military spending and green capital. One reason could be that these countries are more supportive of environmental awareness programmes. Since militarization leads to environmental degradation (Renner 1991; Clark et al. 2010; Lin et al. 2015; Bildirici 2017),

when military spending is increased, the government may increase spending on green capital as well. According to military Keynesianism, increasing military spending can generate aggregate demand, which in turn stimulates welfare expenditures, demonstrating a complementary link between military and welfare spending.

## 4. Conclusions and Policy Suggestions

This study started its venture with the objectives of examining four hypotheses: whether long-run relations exist between military spending and green capital, whether there are short-run dynamics, whether there are causal interplays between the two and whether the former crowds-in or crowds-out the latter. The results show that there is the existence of long-run relations between the two for the majority of the countries and military spending makes a cause to green capital in the long run, justifying the error correction results. However, there are a few countries on the list where the study observes causal interplay from military heads to green capital heads. Finally, the study finds that the militarization practices crowd-out the green capital formation in eight countries and the opposite outcome, the crowding-in effects, in twelve countries. The countries falling under the crowding-out category are still not aware of environmental degradation while those of the countries falling in the crowding-in category, are well aware of environmental degradation.

The study could provide better results if we could consider similar investigations for the panels of the countries separately, for the developed countries' group, developing countries' group, and total 20 countries group to have a more robust result. This is a limitation of the present study. We aim to exercise the left-out part, the panel data analysis, in a future program.

It is thus the task of the policy makers of the concerned countries to think of the environmental issues seriously by deploying more funds upon green investments by substituting the military expenses. Further, if the countries think of the importance of militarization still for national security, they should generate additional funds for conservation of the nature. Hence, the debate of 'gun vs. butter' still stands, and the policy makers should alter the debate into proposition in the way of 'gun and butter' framework.

**Author Contributions:** Conceptualization, R.C.D.; methodology, R.C.D.; validation, R.C.D.; investigation, R.C.D., I.H.; data curation, I.H.; writing—original draft preparation, R.C.D.; writing—review and editing, R.C.D., I.H.; visualization, R.C.D.; supervision, R.C.D.; funding acquisition, R.C.D. All authors have read and agreed to the published version of the manuscript.

**Funding:** This research received no external funding.

**Institutional Review Board Statement:** Not applicable.

**Informed Consent Statement:** Not applicable.

**Data Availability Statement:** Data of military expenditures are taken from the Stockholm International Peace Research Institute (SIPRI) (https://www.sipri.org/, accessed on 1 February 2023) and the data on green capital is obtained from OECD (https://www.oecd.org/, accessed on 1 February 2023).

**Conflicts of Interest:** The authors disclose that they did not face any conflict of interest while developing the full manuscript in the said topic. Further they disclose that they did not have any funding source behind the completion of the work.

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
