# Peer review of "Relationships between Military Spending and Green Capital Formation: Complementary or Substitutes?"

_socsci, doi:10.3390/socsci12100571_

Round 1
Reviewer 1 Report
The article raises the extremely important problem of the relationship between military spending and the creation of green capital. The presented research results are interesting, but the text needs to be corrected:
1. Abstract: abstract is too long, according to the publisher's recommendations it should not exceed 200 words. In addition, the aim of the research should be clearly formulated. Hypothesis testing is not an aim in itself. The aim may be identification, indication, assessment etc..
2. Introduction - Second paragraph, lines 50-56: these sentences are better written in the past tense, because they are about 40-50 years ago. Last paragraph, lines 70-71, 81: writing about literature requires references - which researchers are referenced.
3. Theoretical background - this section should be part of the Introduction without additional title. References are needed.
4. Review of Related Literature – this section should be part of the Introduction.
5. Research Gaps and Questions – this section should be part of the Introduction. Line 201-202: how do you know there is no article? Research questions should be formulated starting with the words how, what, etc. As it stands, only yes or no answers can be given, which makes the questions sound like hypotheses.
6. Hypotheses of the Study – this section should be presented along with the research methods. The first hypothesis is better divided into two hypotheses. The second hypothesis needs to be formulated differently - avoiding negation in conjunction with "as well as".
7. Variables and the Data Source: this section should be part of the Materials and Methods section, which is missing from the article. There should be a separate section where, together with data sources, it is worth presenting hypotheses and how they were tested.
8. Graphical presentation of data - better moved to Results without additional title.
9. Results and Discussion - from this section, research methods should be separated and moved to Materials and Method (formulas and tests), leaving only the results (tables, graphs and their descriptions) and discussion. It's better to remove the extra subtitles. Line 326 - should be "Table 3". To the section add findings, limitations and future research directions may also be mentioned.
It is worth adapting the sections of the article to the publisher's requirements: https://www.mdpi.com/journal/socsci/instructions
Good luck!
Author Response
Thank you for the comments and suggestions. Please find the attached file on Authors' Response.

Reviewer 2 Report
The following changes are suggested before publishing the article: (1) include the main objective of the study in the abstract (the objective cannot be to verify the research hypotheses); (2) re-edit the introduction (research gaps should be indicated before the main objective; research hypotheses and specific objectives should also be presented in this section; the description of the different sections of the article is missing); (3) replace figures 1-2 with tables (in their current form they are unreadable); (4) change the formatting of Table 1 (change the values to millions of dollars; give the values of the coefficients max. to hundredths); (5) re-edit the conclusion: conclusions, reference to the purpose and research hypotheses; suggestions for policy; limitations of the study; future research directions.
It is recommended to reread the article for spelling errors.
Author Response
Thank you for the suggestions and comments. Please find the attached file on Authors' Response.

Round 2
Reviewer 1 Report
Thank you for considering my tips. After the corrections, the article is more readable and will certainly enrich the journal's publications.
Thank You!